# Synthesized *Bis*-Triphenyl Phosphonium-Based Nano Vesicles Have Potent and Selective Antibacterial Effects on Several Clinically Relevant Superbugs

**DOI:** 10.3390/nano14161351

**Published:** 2024-08-15

**Authors:** Silvana Alfei, Guendalina Zuccari, Francesca Bacchetti, Carola Torazza, Marco Milanese, Carlo Siciliano, Constantinos M. Athanassopoulos, Gabriella Piatti, Anna Maria Schito

**Affiliations:** 1Department of Pharmacy, University of Genoa, Viale Cembrano, 16148 Genoa, Italy; francesca.bacchetti@edu.unige.it (F.B.); carola.torazza@unige.it (C.T.); marco.milanese@unige.it (M.M.); 2Laboratory of Experimental Therapies in Oncology, IRCCS Istituto Giannina Gaslini, Via G. Gaslini 5, 16147 Genoa, Italy; 3IRCCS, Ospedale Policlinico San Martino, 16132 Genova, Italy; 4Department of Pharmacy, Health and Nutritional Sciences, University of Calabria, 87036 Rende, Italy; carlo.siciliano@unical.it; 5Department of Chemistry, University of Patras, University Campus Rio Achaias, 26504 Rio, Greece; kath@chemistry.upatras.gr; 6Department of Surgical Sciences and Integrated Diagnostics (DISC), University of Genoa, Viale Benedetto XV, 6, 16132 Genova, Italy; gabriella.piatti@unige.it (G.P.); amschito@unige.it (A.M.S.)

**Keywords:** multidrug resistant bacteria, triphenyl phosphonium salts, mitochondria target bola-amphiphiles, membranes permeabilization, MICs determination, time-killing experiments, cytotoxicity studies

## Abstract

The increasing emergence of multidrug-resistant (MDR) pathogens due to antibiotic misuse translates into obstinate infections with high morbidity and high-cost hospitalizations. To oppose these MDR superbugs, new antimicrobial options are necessary. Although both quaternary ammonium salts (QASs) and phosphonium salts (QPSs) possess antimicrobial effects, QPSs have been studied to a lesser extent. Recently, we successfully reported the bacteriostatic and cytotoxic effects of a triphenyl phosphonium salt against MDR isolates of the *Enterococcus* and *Staphylococcus* genera. Here, aiming at finding new antibacterial devices possibly active toward a broader spectrum of clinically relevant bacteria responsible for severe human infections, we synthesized a water-soluble, sterically hindered quaternary phosphonium salt (BPPB). It encompasses two triphenyl phosphonium groups linked by a C12 alkyl chain, thus embodying the characteristics of molecules known as bola-amphiphiles. BPPB was characterized by ATR-FTIR, NMR, and UV spectroscopy, FIA-MS (ESI), elemental analysis, and potentiometric titrations. Optical and DLS analyses evidenced BPPB tendency to self-forming spherical vesicles of 45 nm (DLS) in dilute solution, tending to form larger aggregates in concentrate solution (DLS and optical microscope), having a positive zeta potential (+18 mV). The antibacterial effects of BPPB were, for the first time, assessed against fifty clinical isolates of both Gram-positive and Gram-negative species. Excellent antibacterial effects were observed for all strains tested, involving all the most concerning species included in ESKAPE bacteria. The lowest MICs were 0.250 µg/mL, while the highest ones (32 µg/mL) were observed for MDR Gram-negative metallo-β-lactamase-producing bacteria and/or species resistant also to colistin, carbapenems, cefiderocol, and therefore intractable with currently available antibiotics. Moreover, when administered to HepG2 human hepatic and Cos-7 monkey kidney cell lines, BPPB showed selectivity indices > 10 for all Gram-positive isolates and for clinically relevant Gram-negative superbugs such as those of *E. coli* species, thus being very promising for clinical development.

## 1. Introduction

Multidrug resistant (MDR) bacteria, such as ESKAPE pathogens, being ESKAPE the acronym for Enterococcus faecium, Staphylococcus aureus, Klebsiella pneumoniae, Acinetobacter baumannii, Pseudomonas aeruginosa, and Enterobacter species, are an irrepressible worldwide threat to both humans and animals and a global public health problem [1]. These MDR pathogens are responsible for difficult-to-treat infections, with long-term hospitalization, high management costs, and cause approximately 35,000 deaths per year in the United States [2]. Such a global antibiotic resistance crisis could be limited by reducing their misuse in medicine, agriculture, and the environment [3] as well as by implementing an efficient infection control strategy to prevent the spread of contagions [2,4,5]. Anyway, under biofilm conditions, an even higher dosage of antibiotics is necessary, which paradoxically drives to further bacterial resistance [3].

Alternative treatments can improve the efficacy of existing antibiotics, such as the use of metabolites (L-alanine and glucose) capable of modifying bacterial metabolism. Nanomaterial-based delivery systems could improve target delivery, thus reducing the amount of antibiotics necessary to counteract infections and limiting the further emergence of resistance [6]. Among nanomaterials as alternatives to no longer functioning antibiotics, gold (Au) and silver (Ag) nanoparticles (AuNPs and AgNPs), as well as hybrid systems based on combining metallic NPs with inorganic and organic materials, have been widely and successfully used in biomedicine as antimicrobial agents [7]. Cefiderocol from cephalosporins, omadacycline from tetracycline, and vancomycin A from vancomycin are new agents developed based on already existing classes of antibiotics [6], but the exclusive discovery of novel agents is still inadequate [3]. New antibacterial drugs are searched for by exploring peculiar sources, such as soil actinomycetes, insects, marine samples, or microbiota, as well as by screening small-molecule libraries [6]. A new lead antibiotic was identified by screening 167,405 compounds, which demonstrated inhibitory effects against methicillin-resistant *S. aureus* (MRSA) and vancomycin-resistant *S. epidermidis* (VRE), but no effect was observed against Gram-negative bacterial pathogens, which remain the most concerning [8].

An entirely novel treatment with no tendency to develop resistance is represented by antimicrobial oxidative therapy (AOT), which exploits the action of reactive oxygen species (ROS) induced by different methods [3,9].

ROS have demonstrated significant *in vitro* and *in vivo* antimicrobial properties against a wide spectrum of Gram-positive, Gram-negative organisms, fungi, and parasites, including MDR isolates and biofilm-producing pathogens [10]. To date, the use of ROS represents a clinically approved therapeutic approach only for topical use on the skin, mucosal membranes, or internal tissue and to treat infected wounds, including chronic wounds, superficial and acute wounds, burns, surgical wounds, cuts, abrasions, and diabetic foot ulcers, but it is not suitable for systemic use [11,12]. Further clinical investigations are required to determine the possible toxic effects of ROS on mammalian cells [9].

Synthetic cationic surfactants, including quaternary ammonium salt (QAS) derivatives of benzalkonium chloride (CITROSIL^®^), have been studied since 1930 [13]. QASs demonstrated considerable broad-spectrum potency and biocidal effects against Gram-positive and Gram-negative bacteria, some fungi, parasites, and even enveloped lipophilic viruses [14,15]. Their antimicrobial activity depends mainly on their charge, the number of cationic centers, the nature of the counter-ion, the length and number of alkyl substituents, and the possible presence of aromatic groups [1,16]. As recently reported, QASs can include single-chain QASs, double-chain QASs, heterocyclic QASs, gemini QASs, and QASs with more than two positively charged moieties, depending on the number and structure of their long carbon chains and on the number and typology of their cationic heads [13].

Although some additional possible mechanisms for these biocides have been proposed [17], they generally disrupt the cell membranes of microorganisms [18,19] Specifically, after electrostatic interaction with the external negative constituents of bacteria, by intercalating into the membranes, altering their normal structure, and causing depolarization, QASs succeed in modifying membrane permeability by pore formation, thus determining the loss of ions, enzymes, and coenzymes. The biosynthetic activity of bacteria is irreversibly compromised, thus prompting their death [14,20,21]. Due to these actions, QASs have been demonstrated to be active against worrying MDR strains such as *E. faecium*, *S. aureus*, *K. pneumoniae*, *A. baumannii,* and *P. aeruginosa* [22].

Unfortunately, the non-specific antimicrobial mechanism of QASs and their noteworthy cytotoxicity and hemolytic toxicity in eukaryotic cells [14,15,23] impede their oral and systemic administration in vivo, thus limiting their use as surface disinfectants and antibacterial devices for topical treatment [18].

A more recently studied class of compounds, structurally similar to QASs and with similar biological properties, consists of quaternary phosphonium salts (QPSs) [24,25,26]. QPSs have shown higher biological activities than QASs, including antibacterial and antitumor effects, because of the difference in electronegativity between phosphorus and carbon atoms [21,27,28]. On the contrary, a minor cytotoxicity was observed for QPSs with respect to structurally analogous QASs [28,29]. QPSs with C8-C11 alkyl chains have demonstrated significant and discerning antibacterial effects especially against Gram-positive species [30,31].

Seddon et al. demonstrated that the antimicrobial properties of *n*-hexyl-(alkyl)-phosphonium ionic liquids (PILs) were based on the alkyl substituent length and depended on the type of anion [31]. In the development of strategies to regulate the biological properties of QPSs through structural tuning, the peculiar structure of sterically hindered phosphonium salts (HPSs), where the charged phosphorus atoms are shielded, and the charge of their cation(s) has a particular distribution, supports the hypothesis that such compounds could possess distinctive biological properties and different mechanisms of interaction with living cells, with a possible reduced hemolytic toxicity and cytotoxicity [1].

Ermolaev et al. studied the structure-activity relationships (SAR) of tri-*tert*-butyl phosphonium salts having form C1 to C20 alkyl chains and bromide, iodide, or BF_4_^−^ anions [1].

When tested on *S. aureus*, *Bacillus cereus*, *E. faecalis*, *E. coli*, *P. aeruginosa*, as well as MRSA isolates resistant to fluoroquinolones and/or β-lactams and some fungi, compounds with C11 to C18 chains demonstrated from good to excellent selective antimicrobial effects [1]. Hindered QPSs based on triphenylphosphine have been studied to an even greater extent, thus evidencing a broad spectrum of biological activities and antifouling properties, as recently reported by us [13]. *Bis* (triphenyl phosphonium) salts (BTPPSs), which include bola-amphiphiles (a class of surfactants featuring one or more hydrophobic chains that connect two identical or different hydrophilic headgroups), have shown sterilizing properties [32,33], antibacterial activity higher than that of mono triphenyl phosphonium salts, and low levels of cytotoxicity [34,35]. This major antibacterial activity and low cytotoxicity could be attributed to the presence of more than one cationic moiety and to the “soft” cationic nature of the headgroups, characterized by extended delocalization of the positive charge on the phenyl rings, respectively [36]. These substances possess a greater ability to interact with and cross membranes with highly negative potentials, such as those of bacteria, damaging them and facilitating their entry into the cell [36]. Due to the difference in negative electrical membrane potentials between bacterial and eukaryotic cells, these compounds can selectively interact with bacterial membranes and accumulate within pathogen cells compared to those of mammals [34]. Once inside the bacterial cell, BTPPSs can detrimentally interact with other negative constituents, including genomic and plasmid DNA, thus causing cell death [30]. With these premises, aiming at finding new antibacterial devices, finalized at counteracting human and animal infections sustained by MDR Gram-positive and Gram-negative strains, no longer treatable with available antibiotics, we synthesized and characterized the water-soluble sterically hindered quaternary phosphonium salt (BPPB) shown in Figure 1.

BPPB was characterized by ATR-FTIR, NMR, UV spectroscopy, FIA-GC (ESI), and elemental analysis to confirm its structure. Potentiometric titrations, optical microscopy, and DLS analyses were also carried out. For the first time, the antibacterial effects of a bola-amphiphile molecule, such as BPPB, were assessed against a multitude (50 strains) of clinically relevant superbugs of different genera of both Gram-positive and Gram-negative species. All isolates used here had difficult-to-manage patterns of resistance, thus being responsible for almost intractable human and animal infections. MICs determinations were carried out with BPPB on all isolates considered in this study, while 24-h time-killing experiments were performed on different strains of *E. coli* and MRSA, as representative isolates of Gram-positive and Gram-negative species. Moreover, for the first time, cytotoxicity experiments were also performed, using HepG2 human hepatic and Cos-7 monkey kidney cells as *in vitro* models of eukaryotic cells, to establish the potential clinical application of BPPB.

## 2. Materials and Methods

### 2.1. Chemicals and Instruments

All reagents and solvents were from Merck (formerly Sigma-Aldrich, Darmstadt, Germany) and were purified by standard procedures. The organic solutions were evaporated using a rotatory evaporator operating at a reduced pressure of about 10–20 mmHg. The melting range of BPPB was determined on a 360 D melting point device with a resolution of 0.1 °C (MICROTECH S.R.L., Pozzuoli, Naples, Italy). The melting point was uncorrected. Attenuated total reflectance (ATR) Fourier transform infrared (FTIR) analyses were carried out using the same instrument as previously reported [37]. ^1^H, ^13^C, and ^31^P NMR spectra of BPPB were acquired on a Jeol 400 MHz spectrometer (JEOL USA, Inc., Peabody, MA, USA) at 400, 100, and 162 MHz, respectively. Standard ^13^C NMR spectra were reported. Chemical shifts were reported in ppm (parts per million) units relative to the internal standard tetramethylsilane (TMS = 0.00 ppm)., and the splitting patterns were described as follows: s (singlet), d (doublet), t (triplet), q (quartet), m (multiplet), and br (broad signal). FIA-MS experiments were performed on a linear ion trap mass spectrometer (LXQ Thermo Finningan, Thermofisher, Milan, Italy, PR) equipped with a heated electrospray ionization (HESI) source. Operating conditions of the HESI were as follows: source temperature: 50 °C; ion spray voltage = +4.0 kV; sheath gas = 5 (arbitrary scale); capillary temperature = 275 °C. Methanolic solutions of the samples (1 × 10^−5^ M) were infused via a syringe pump at a flow rate of 3–5 μL/min. Potentiometric titrations were carried out using a Hanna Micro-processor Bench pH Meter (Hanna Instruments Italia srl, Ronchi di Villafranca Padovana, Padova, Italy) in mV mode, which was calibrated using standard solutions at pH = 4, 7, and 10 before titrations. Elemental analyses were carried out using an Elemental Analyzer (Fison Instruments Ltd., Farnborough, Hampshire, UK). Thin layer chromatography (TLC) was performed using aluminum-backed silica gel plates (Merck DC-Alufolien Kieselgel 60 F254, Merck, Washington, DC, USA), and detection of spots was made by UV light (254 nm) using a Handheld UV Lamp, LW/SW, 6W, UVGL-58 (Science Company^®^, Lakewood, CO, USA).

### 2.2. Synthesis of 1,1-(1,12-Dodecanediyl)bis[1,1,1]-triphenylphosphonium di-Bromide (BPPB)

The following Figure 2 shows the numbered structure of BPPB which will be useful to readers for understanding the peaks assignation in the NMR peaks lists reported in this section.

A previously reported synthetic procedure was here substantially modified and optimized [36]. 1,12-dibromo-dodecane (1.00 g, 0.003 moles) and triphenylphosphine (TPP) (1.60 g, 0.006 moles) were dissolved in 30 mL ethanol (EtOH). The reaction mixture was stirred and refluxed for 48 h. Then, the solvent was removed under vacuum to obtain a yellowish resin that was recrystallized by q.b. dichloromethane (DCM)/di-isopropyl ether (DIPE) as a coupled solvent/non-solvent to obtain a white precipitate which was separated by filtration, washed twice with diethyl ether (Et_2_O), and brought to a constant weight at reduced pressure. The final product was obtained as hygroscopic low-melting point white foam (2.18 g, 0.0026 moles, 85.2%). Melting point (M.p.) < 70 °C. ATR-FTIR (ν, cm^−1^): 3052, 3007 (C-H stretching phenyl rings), 2923 (CH_2_), 2852 (CH_2_), 1586, 1484 (C=C stretching), 1436, 995 (aromatic C-P), 689 (aliphatic C-P). ^1^H-NMR (400 MHz, CDCl_3_) ppm: 1.16 (m, 4H, [6,9] CH_2_), 1.22 (m, 4H, [5,10] CH_2_), 1.23 (m, 4H, [7,8] CH_2_), 1.50 (m, 4H, [4,11] CH_2_), 1.60 (m, 4H, [3,12] CH_2_), 3.62 (dt, 4H, [2,13] CH_2_, *J*_CH2(2,13)-P_ = 13.0 Hz, *J* = 6.20 Hz), 7.63 (m, 12H, *meta* ArH), 7.67 (m, 6H, *para* ArH), 7.95 (m, 12H, *orto* ArH). ^13^C-NMR (100 MHz, CDCl_3_) ppm: 21.94 (2d, [3,12] C); 22.26 (2d, [2,13] C, *J*_C(2,13)-P_ = 50.10 Hz); 28.60 (2s, [7,8] C); 29.39 (2d, [6,9] C); 29.55 (2d, [5,10] C); 30.17 (2d, [4,11] C); 118.32 (6d, ArC-P, *J*_ArC-P_ = 86.0 Hz); 130.09 (12d, ArCH*orto*, *J*_ArCH*orto*-P_ = 12.5 Hz); 133.53 (12d, ArCH*meta*, *J*_ArCH*meta*-P_ = 10.2 Hz); 134.72 (6s, ArCH*para*). ^31^P-NMR (162 MHz, CDCl_3_) ppm: 26.72 (s, P nuclei 1.14). FIA-MS-(ESI): 346.25 *m*/*z* [C_48_H_54_P_2_]^2+^. Anal. Calcd. for C_48_H_54_P_2_Br_2_: C, 64.61; H, 6.38; P, 7.26. Found: C, 64.63; H, 6.41; P, 7.30.

### 2.3. UV-Vis Analyses

The UV-Vis spectrum of BPPB was acquired in water at room temperature. Subsequently, BPPB (6.8 mg) was dissolved in 0.5 mL water by gentle heating, obtaining a clear solution of 13.6 mg/mL (15.9 mM). Then, 220 µL was withdrawn and diluted to 50 mL to obtain a solution with a concentration similar to that reported by Cecacci et al. [36]. The obtained solution was analyzed using an Agilent Cary 100 UV/Vis Spectrophotometer (Agilent Technologies Italia S.p.A., Milan, Italy). Analyses were performed in triplicate and the image reported in the Results and Discussion section is representative of the acquisition, which presented new results that were different from those previously reported [36].

### 2.4. Potentiometric Titration of BPPB

Potentiometric titration of BPPB was carried out in a mixture of anhydrous acetic acid (AcOH) and acetic anhydride (Ac_2_O) 30:70 (*v*/*v*)), with HClO_4_, using a procedure previously described by us for the volumetric titration of primary ammonium salts, with slight modifications [38,39,40]. The protocol was first described by Pifer and Wollish for titrating quaternized ammonium salts [41]. Succinctly, an exacted weighted sample of BPPB (280.0 mg) was dissolved in AcOH/Ac_2_O 30/70 (5 mL), treated with 2–4 mL of a solution of mercury acetate (1.5 g) in AcOH (25 mL), and titrated with a standardized 0.1 N solution of HClO_4_ in AcOH/Ac_2_O, prepared as described in the following section, using potentiometric endpoint detection. The titrations were performed under efficient stirring with a magnetic stirrer at room temperature (25 ± 2 °C). Millivolts were measured every 0.5 mL up to 6 mL and every 0.1 mL in the vicinity of the calculated endpoint up to the addition of 8 mL 0.1 N HClO_4_. Titrations were performed in triplicate, and the measurements are reported as mean ± SD.

#### Preparation of 0.1 M Perchloric Acid Volumetric Solution

The 0.1 M perchloric acid volumetric solution was prepared by diluting 8.5 mL of 70–73 wt% perchloric acid with 900 mL of anhydrous acetic acid and 30 mL of acetic anhydride and then diluting to 1000 mL with anhydrous acetic acid. Perchloric acid was standardized by titration with potassium hydrogen phthalate.

### 2.5. Optical Microscopy Analyses

The morphology of BPPB as a water suspension was investigated via optical microscopy (OM) analysis. In the performed experiments, 6.8 mg of solid BPPB was dissolved in 0.5 mL water by gentle heating, obtaining a clear solution of 13.6 mg/mL (15.9 mM). The fine suspension obtained on cooling was observed using a Leica DM750 optical microscope (Leica Italy, Milan, Italy) equipped with 40× and 100× objectives. The camera used for image capture was a Leica ICC50W (Leica Italy, Milan, Italy). All images were processed using LAS EZ 3.4.0. software (Leica Italy, Milan, Italy).

### 2.6. Dynamic Light Scattering (DLS) Analysis

The particle size (nm) intended as hydrodynamic diameter distribution, polydispersity index (PDI), and zeta potential (ζ-p) (mV) of BPPB were measured at 25 °C, at a scattering angle of 90° in m-Q water using a Malvern Nano ZS90 light scattering apparatus (Malvern Instruments Ltd., Worcestershire, UK).

A solution of BPPB 10 mM in m-Q water (312.4 Kcps) was diluted 1:2 to a final concentration of 5 mM (8.4 kcps) and analyzed. The ζ-p value of BPPB was recorded using the same apparatus at a count rate of 20–59 kcps. The results of the experiments are presented as the mean of 3 independent determinations, made of 10 runs (particle size) or 12 runs (ζ-p), each one ± SD. Intensity-based results have been reported to express particle size distribution.

### 2.7. Microbiology

#### 2.7.1. Microorganisms

A total of 50 isolates belonging to a collection of MDR Gram-positive and Gram-negative species of the University of Genova were used in this study. All were clinical strains isolated from human specimens and identified using VITEK^®^ 2 (Biomerieux, Firenze, Italy) or matrix-assisted laser desorption/ionization time-of-flight (MALDI-TOF) mass spectrometric technique (Biomerieux, Firenze, Italy). The 50 MDR isolates included 22 Gram-positive and 28 Gram-negative bacteria of different genera. Among bacteria of Gram-positive species, 8 were enterococci (4 *E. faecalis* and 4 *E. faecium*), while 14 were staphylococci (8 *S. aureus* and 6 *S. epidermidis*). All enterococci were MDR isolates with resistance to vancomycin (VRE), while all staphylococci were MDR strains with resistance to methicillin (MRSA and MRSE). Additionally, three MRSE strains (*S. epidermidis*) also displayed resistance to linezolid. Gram-negative species included non-fermenting isolates such as *P. aeruginosa* (4 isolates), *A. baumannii* (5 strains), and *Stenotrophomonas maltophylia* (5 isolates), as well as *Enterobacteriaceae*, such as *E. coli* (5 isolates), *K. pneumoniae* (5 strains) *Klebsiella aerogenes* (1 isolate), 2 isolates of *Enterobacter cloacae* and 1 isolate of *Enterobacter hormaechei*. Among four *P. aeruginosa*, three were isolated from cystic fibrosis patients and were resistant to carbapenems. Two of them were also pyoverdine and pyocyanin producers, while one was ceftazidime-avibactam (CAZAVI) resistant. The fourth *P. aeruginosa* was an MDR isolate that was also resistant to colistin. All five *A. baumannii* were carbapenems resistant MDR isolates, while the five *S. maltophylia* were MDR strains of which one was also resistant to cotrimoxazole (trimethoprim and sulfamethoxazole). Except for one *E. coli* that was fully sensitive, all the residual four *E. coli* and all the five *K. pneumoniae* were resistant to carbapenems. Four *K. pneumoniae* were KPC producers, one of which was also colistin-resistant, while one was a metallo-β-lactamase VIM-1 producer with resistance to the recently approved cefiderocol. Among MDR *E. coli*, two were producers of the New Dheli metallo-β-lactamase (NDM), one of which also displayed resistance to cefiderocol, one was a VIM-1 producer, while the last *E. coli* was a KPC producer isolate. *K. aerogenes* was a KPC producer isolate, *E. cloacae* isolates were VIM-1 producers, while *E. hormaechei* demonstrated resistance also to cefiderocol.

#### 2.7.2. Determination of the MICs

To investigate the antimicrobial activity of BPPB on the described pathogens, their Minimal Inhibitory Concentrations (MICs) were determined by following the microdilution procedures detailed by the European Committee on Antimicrobial Susceptibility Testing EUCAST [42], and reported in our previous works [13].

#### 2.7.3. Killing Curves

Killing curve assays for BPPB were carried out on isolates of *S. aureus* (strains 17, 18, 187, and 195, all MRSA), and *E. coli* (strains 477, 525, and 539), following a previously reported procedure [43]. Experiments were performed over 24 h at a BPPB concentration four times the MIC for all strains.

A mid-logarithmic phase culture was diluted in Mueller–Hinton (MH) broth (Merck, Darmstadt, Germany) (10 mL) containing 4 × MIC of the selected compound to give a final inoculum of 1.0 × 10^5^ CFU/mL. The same inoculum was added to cation-supplemented Mueller–Hinton broth (CSMHB) (Merck, Darmstadt, Germany) as a growth control. The tubes were incubated at 37 °C with constant shaking for 24 h. Samples of 0.20 mL from each tube were removed at 0, 2, 4, 6, and 24 h, diluted appropriately with a 0.9% sodium chloride solution to avoid carryover of BPPB being tested, plated onto MH plates, and incubated for 24 h at 37 °C. The growth controls were run in parallel. The percentage of surviving bacterial cells was determined for each sampling time by comparing the colony counts with those of standard dilutions of the growth control. Results have been expressed as log10 of viable cell numbers (CFU/mL) of surviving bacterial cells over a 24 h period. The bactericidal effect was defined as a 3 log10 decrease in CFU/mL (99.9% killing) of the initial inoculum. All time-kill curve experiments were performed at least in triplicate.

### 2.8. Concentration-Dependent Cytotoxicity Experiments: MTT and LHD Tests

#### 2.8.1. MTT Cytotoxicity Assay

To assess the cytotoxic properties of BPPB, the MTT (3-(4,5-dimethylthiazol-2-yl)-2,5-diphenyltetrazolium bromide) cell proliferation assay [44] (Abcam, Milan, Italy, Cat#ab2011091) was performed, following the manufacturer’s protocol, as previously reported [13]. Briefly, Cos-7 cells (African green monkey kidney fibroblast-like cell line) and HepG2 cells (human liver epithelial-like cell line) were plated at 20,000 cells/well into 96-well plates; the cells were maintained in complete Dulbecco’s Modified Eagle Medium (DMEM; Euroclone, Milan, Italy, Cat# ECM0728L) containing 10% Fetal Bovine Serum (Euroclone, Milan, Italy, Cat# ECS0180L), 1% glutamine (Euroclone, Milan, Italy, Cat# ECB3004D) and 1% Penicillin/Streptomycin (Euroclone, Milan, Italy, Cat# ECB3001D) at 37 °C, in a 5% CO_2_ atmosphere for 24 h, as for MICs determinations. Once the optimal cell confluence was verified, the complete DMEM was replaced with fresh media containing increasing concentrations of bis-phosphonium bromide (range of 0.5–100 µg/mL), and the cells were incubated at 37 °C in 5% CO_2_ for a further 24 h. After that, the media was removed, and the cells were washed with PBS. Aliquots (200 µL) of serum-free medium containing MTT (Merck, Milan, Italy, Cat #M5655; 0.25 mg/mL MTT) were added to each well and incubated at 37 °C for 3h. After removing the medium, 200 µL of DMSO solution (Merck, Milan, Italy, Cat #276855; 0.25 mg/mL MTT) was added to each well and horizontally shaken for 10 min to allow DMSO to solubilize the formazan crystals, allowing the formation of a homogenous solution. The 570 nm wavelength light absorption was then measured spectrophotometrically in each well using the Spectra Max 340 PC (Molecular Devices, San Jose, CA, USA) and converted into OD (optical density) units. The cell survival rate, expressed as cell viability percentage (%), was evaluated based on the experimental outputs of the treated groups vs. the untreated groups (CTR) and was calculated as follows: cell viability (%) = (OD treated cells − OD blank)/(OD untreated cells − OD blank) × 100%.

#### 2.8.2. LDH Cytotoxicity Assay

To assess the cytotoxicity properties of bis-phosphonium bromide, a lactate dehydrogenase (LDH) cytotoxicity assay [45] (Abcam, Milan, Italy, Cat#ab102526) was performed, following the manufacturer’s protocol. Briefly, Cos-7 and HepG2 cells were seeded at 20.000 cells/well into 96-well plates, the cells were maintained in complete Dulbecco’s Modified Eagle Medium (DMEM; Euroclone, Milan, Italy, Cat# ECM0728L) containing 10% Fetal Bovine Serum (Euroclone, Milan, Italy, Cat# ECS0180L), 1% glutamine (Euroclone, Milan, Italy, Cat# ECB3004D) and 1% Penicillin/Streptomycin (Euroclone, Milan, Italy, Cat# ECB3001D) at 37 °C in a 5% CO_2_ atmosphere for 24 h. Once the optimal cell confluence was verified, the complete DMEM was replaced with fresh media containing increasing concentrations of bis-phosphonium bromide (range of 0.5–100 µg/mL), and the cells were incubated at 37 °C in 5% CO_2_ for a further 24 h, as for MICs determinations. The supernatant was collected and immediately transferred to new 96-well flat-bottom enzymatic assay plates. Fifty microliters of LDH assay substrate was added to the medium, and the absorbance measurement output was immediately recorded at 450 nm, using a microplate reader (Spectra Max 340 PC, Molecular Devices, San Jose, CA, USA) set up in kinetic mode, every 2–3 min, for at least 30 min at 37 °C. The positive control outputs were used as the maximum LDH signal. The LDH content (OD at 450 nm) quantified in the culture medium collected from treated or untreated conditions was directly proportional to damaged cells.

#### 2.8.3. Statistics

Graphs and statistics were generated using GraphPad Prism (Version 9, license code GP9-2314983-RATL-05225; 225 Franklin Street. Fl. 26, Boston, MA, USA 02110; RRID:SCR_002798).

## 3. Results and Discussion

### 3.1. Synthesis of 1,1-(1,12-Dodecanediyl)bis[1,1,1]-triphenylphosphonium di-Bromide (BPPB)

With the scope to develop novel antibacterial agents effective against MDR bacteria of both Gram-positive and Gram-negative species, and due to the reported antibacterial effects of QPSs [13,21,27,30,31], and those even superior to *bis*-(triphenyl phosphonium) salts [34,35], we synthesized the dodecyl *bis*-triphenilphosphonium derivative (Figure 1) according to Figure 1.

The C12 chain was selected because among the hindered mono phosphonium bromides reported by Ermolaev et al. [1], those bearing a C12 alkyl chain demonstrated very high antibacterial effects, a very low level of hemotoxicity, and little cytotoxicity to Chang liver cells. The C12 derivative demonstrated the highest selectivity index for bacteria *S. aureus* 209 P and MRSA compared to blood cells, also when used to coat polymer surfaces [1]. Additionally, even if only against environmental bacteria and never against MDR clinical isolates as we made, the sterilizing action of *bis*-phosphonium bola-amphiphilies bearing a C12 chain was reported [32,33]. EtOH was used as the solvent in place of DMF, which was used by Caccacci et al. [36] to facilitate solvent removal and voids to retain residual DMF in the product. The reaction advancement was followed by TLC (AcOEt/Petrol ether 1/1), and it was stopped when the spot of triphenylphosphine (TPP, Rf = 0.25) disappeared. In this eluent mixture, BPPB displayed Rf = 0. The pale yellow resin obtained by removing the solvent under pressure was recrystallized using diclholoromethane (DCM)/di-isopropyl ether (DIPE) to obtain pure BPPB as a low-melting-point white foamy solid with a high tendency to absorb atmospheric humidity and yield comparable [36] or even higher [32,33] than those previously reported. BPPB was, for the first time, characterized by ATR-FTIR (Figure 3). ^1^H NMR (Figure 4), ^13^C-NMR (Figure 5), and for the first time, ^13^C-NMR DEPT 135 and ^31^P-NMR spectra were acquired and furnished as images (Figure 6). FIA-MS experiments, UV analyses, and elemental analyses that had never been provided so far were also carried out. The results further confirmed the BPPB structure, its good degree of purity, and evidenced new unexpected findings.

Particularly, in the FTIR spectrum, the weak bands of the phenyls C-H stretching (3052 and 3007 cm^−1^) were observable, while the bands of the symmetric and asymmetric stretching of C–H of the several methylene groups of the C12 alkyl chain were observable at 2923 and 2852 cm^−1,^ respectively [46]. The band of aliphatic C-P stretching was detected at 689 cm^−1^, while two bands at 1436 and 995 cm^−1^ were observed for aromatic C-P stretching [47]. The stretching vibration bands of aromatic C=C bonds in the phenyl groups were instead visible at 1484 and 1586 cm^−1^ [33,46].

The ^1^H NMR spectrum of BPPB showed five broad multiplets in the range 1.16–1.60 ppm at 1.16, 1.22, 1.23, 1.50, and 1.60 integrable for 4H each and belonging to the C4-C11 (see the numbered structure of BPPB in Figure 2) methylene groups. The signal of the CH_2_-P^+^ groups was instead detected as a double triplet at 3.62 ppm whose multiplicity is due to the spin-spin coupling of protons with both the vicinal methylene group (*J* = 6.20 Hz) and with the phosphorus atom (*J* = 13.0 Hz). Finally, a complex signal made of three multiplets centered at 7.63, 7.67 and 7.95 ppm and integrable for 12H, 6H, and 12H, respectively, was detected in the range 7.61–7.99 ppm due to the proton atoms of the six phenyl rings.

At high fields (21.92–30.24 ppm) in the ^13^C-NMR spectrum of BPPB, six signals due to the C2, C3, C4, C5, C6, C7, C8, C9, C10, and C11 (see the numbered structure of BPPB in Figure 2) methylene groups of the aliphatic chain were detected (Figure 5), which appeared downwards-oriented in the DEPT-135 analysis (Figure 6). Carbon spin-spin coupling with the phosphorous atom was observed for C2, C13 (*J*_C2,13-P_ = 50.1 Hz), while no coupling was observed for other carbon atoms of the alkyl chain. The doublet signals of the quaternary carbon atoms of benzene rings directly bonded to P^+^, which disappeared in the DEPT-135 analysis, were found at 118.32 ppm and showed a coupling constant *J*_C-P_ = 86.00 Hz. Whereas the *orto*- and *meta*-CH carbon atoms gave two doublets at 130.09 ppm (*^o^*CH benzene ring, *^o^J*_C-P_ = 12.50 Hz) and at 133.83 ppm (*^m^*CH benzene ring, *^m^J*_C-P_ = 10.20 Hz). The *para*-CH was finally found at 134.72 ppm and showed an insignificant CP coupling. The ^31^P NMR spectrum evidenced only the singlet of the equivalent phosphorous atoms at 26.72 ppm, which is in accordance with the chemical shift reported for similar TPP derivatives [34,35,48,49,50].

### 3.2. UV-Vis Analyses

In the literature, the UV-Vis spectra in water of four *bis*-triphenyl phosphonium bola-amphiphiles having C12, C16, C20, and C30 alkyl chains have been reported as acquired water and measuring the absorbance (Abs) in the range of 250–320 nm [36], detecting three peaks of absorbance whose wavelengths have not been reported [36]. Here, the UV spectrum of BPPB was acquired in water at room temperature in the range 190–390 nm. Figure 7 shows the UV spectrum obtained.

Using a BPPB solution at the same concentration reported by Ceccacci et al. [36], but acquiring the spectrum in a larger range, it was possible for us to detect two very high peaks of absorbance under 250 nm (205 and 224 nm, Abs = 3.3 and 2.7), not reported previously, because of the too high cut-off adopted. The spectrum reported by Ceccacci et al., where three peaks of absorbance were detected in the range of 250–290 nm (Abs of about 0.3), in our spectrum is indicated by the blue circle. In fact, by cutting and rescaling our spectrum, as made by Ceccacci et al. (Figure 8), a UV spectrum identical to that previously reported was obtained [36].

### 3.3. Potentiometric Titration of BPPB in Non-Aqueous Medium

The molecular weight (MW) and structure of BPPB were further confirmed by titrating its phosphonium groups. Using this method, the P^+^ equivalents contained in an exactly weighted sample of BPPB were experimentally determined. By comparing the number of P^+^ equivalent measures with those calculated according to the MW required by the formula of BPPB, we would have validated its mass and structure. In this regard, the emergence of non-aqueous titrations in the middle of the last century has enabled the possibility to determine both weak acids and bases that are not measurable in aqueous media [51,52,53]. Especially, the titration of weak basic drugs with perchloric acid in a glacial acetic acid medium is widely used. Titration in the acetic anhydride/acetic acid mixture enabled direct non-aqueous titration of halide salts (mainly hydrochlorides) of organic bases and quaternary ammonium salts. By adding mercury (II) acetate reagent to the quaternary ammonium salt solution, stable mercury (II) halide complexes and free acetate ions (equivalent to the base) are formed, which can be titrated with perchloric acid [41,54]. On these considerations, we carried out the potentiometric titration of BPPB in a mixture of anhydrous acetic acid (AcOH) and acetic anhydride (Ac_2_O) 30:70 (*v*/*v*)), with 0.1 N HClO_4_, performing a slightly modified procedure previously described by us for the volumetric titration of ammonium salts [38,39,40]. In brief, we reformed the method reported by Pifer and Wollish, who used this procedure to titrate quaternized ammonium salts [41], to examine BPPB. By plotting the measured mV values vs. the volumes of 0.1 N HClO_4_ solution added, we obtained the titration curves of BPPB (Figure 9) and the related first derivative (FD) curves.

The maximum FD represents the titration endpoint, which allowed us to determine the volumes of titrating solution needed to titrate the phosphonium groups of our sample and then their P^+^ equivalents. Table 1 reports the experimental details of titrations, the calculated P^+^ equivalents for BPPB according to its molecular weight (MW = 852.70), the experimentally determined P^+^ equivalents obtained by titrations, the experimental MW, the residuals, and the percentage error (%).

The experimental MW was in perfect agreement with the calculated value, with an error (%) of 0.5%, thus further confirming the structure of BPPB.

### 3.4. Optical Microscopy Results

With the aim of using them as mitochondria-targeted compounds in liposome formulations, Ceccacci et al. prepared and investigated the colloidal behavior of four single-chain bola-amphiphiles compounds, having chains of different lengths linking two triphenyl phosphonium headgroups as BPPB [36]. The authors demonstrated that the studied bola-amphiphiles, upon dispersion in aqueous solution at a concentration higher than their critical aggregative concentration (CAC), were able to spontaneously self-assemble into vesicles independent of the length of the hydrophobic spacer [36]. Particularly, the CAC for the compound having a C12 chain as BPPB was 3.6 mM and for DLS experiments, they prepared solutions of 10 mM. The results showed the presence of two or more dimensional families. The authors assumed that particles with larger dimensions were probably derived from the aggregation of smaller vesicles (about 20 nm) [36]. In this regard, before performing DLS analyses, we investigated the possible capability of BPPB to self-form vesicular aggregates in water solution using optical microscopy. Briefly, we prepared a water dispersion of 15.9 mM, which was clear under gentle heating but opaque on cooling, probably due to the formation of aggregates larger than those present in the clear solution, with a concentration 4.4-fold higher than the CAC of BPPB. The suspension was examined with a 40× (Figure 10a) and 100× (Figure 10b) objective observing spherical polydispersed vesicles.

Due to the high concentration of BPPB, very large aggregates (10–24 µm) were clearly visible using a 40× objective, unequivocally formed by the aggregation of smaller ones (4 µm). Smaller spherical particles were also observed in the background. Using the 100× objective, it was possible to detect much smaller vesicles of 1.58 ± 0.13 µm on a background, evidencing the larger aggregates previously observed in Figure 10a. We are aware that optical microscopy is not a precise method to evaluate the morphological and topographical characteristics of particles because observations are strictly limited by the focus distance, and light can be scattered differently in areas of different densities and geometries. Such a technique has been used only to obtain preliminary results, which confirmed the reported capability of compounds, such as BPPB, to form spherical vesicles of different sizes, which sometimes coexist with aggregates of complex architectures [36]. An optical micrograph was acquired with a 4 × objective on solid BPPB, which showed an amorphous mass with evident traces of liquid despite having been stored properly sealed with parafilm, thus confirming the high hygroscopicity of BPPB (Appendix A).

### 3.5. Particle Size, ζ-p, and PDI of BPPB

The hydrodynamic size (diameter) (Z-AVE, nm) and polydispersity index (PDI) of BPPB vesicles were determined by DLS analysis to assess the dimensions of particles and how much their distribution could be uniform. To this end, BPPB solutions 5 mM filtered using a 0.22-micron filter were used. Additionally, ζ-p measurements were carried out on the same solution to determine their surface charge. Even if characterized by different sizes, preliminary investigations made using BPPB solutions 10 mM evidenced the presence of 2-dimensional families, as previously reported [36]. Precisely, a small dimensional family made of particles of about 379 nm was detected, while the main family was made of very large particles of 1.6 µm (Appendix A), as observed with the optical microscope (Figure 10b). As previously reported, these large particles are derived from the aggregation of smaller vesicles, probably promoted by high concentrations [36]. Upon 1:2 dilution, inverted results were obtained, and the main dimensional family was that with a particle size of 379 nm (Appendix A), reproducing a process similar to that described by Ceccacci et al. [36]. Upon filtration to disrupt residual aggregates, a main dimensional family made of small vesicles (<50 nm) was observed, as shown in Figure 11a, which reports a representative particle size distribution. Nevertheless, in other determinations, other small dimensional families, including even smaller vesicles and larger aggregates, were detected, as shown in Appendix A. The variety of aggregates and vesicles found in DLS analysis is a sign of the well-known bolalipid’s polymorphism [36]. Table 2 reports the average size (Z-AVE, nm) ± SD obtained by three measurements made of ten runs each one, as well as the mean value of PDI ± SD. Figure 11b shows a representative Zeta-potential distribution, while in Table 2 the mean ± SD of the values of ζ-p obtained by three measurements made of 12 runs each one has been included.

As evidenced by the DPI value (0.557) reported in Table 2, polydispersed nanosized vesicles of 45 nm were self-formed by BPPB in an aqueous solution, with a positive ζ-p of +18 mV. As reported by Ceccacci et al., these findings could appear unusual since, generally, bola-amphiphiles with chain spacers of C8-C12 should form vesicles only when a double alkyl chain is present [55]. However, nanosized vesicles have also been observed for bola-amphiphiles with a single chain when a planar cationic headgroup, such as urocanic [56], quinolinium [57], and tri phenyl phosphonium moieties [36], as in our case, are present. This phenomenon occurs because self-assembly properties are strongly dependent on the complex interplay of non-covalent interactions (ionic, hydrophobic, and π-π) inside the aggregate. In this regard, the π-π stacking between the three aromatic rings on polar heads of BPPB was crucial for the final aggregate morphology. Due to the morphology and size of BPPB and in sight of its possible future clinical administration as an antibacterial agent, it must be considered that the size of nanoparticles strongly influences their distribution, cytotoxicity, and targeting ability [37,58,59]. Generally, biomedical applications require sizes lower than 200 nm, with an optimal size of 100–200 nm [37]. Due to the small size of its particles and its hydrosolubility, BPPB could be suitable for systemic administration. Additionally, very small particles, such as those of BPPB, have a larger surface-area-to-volume ratio, are more effective and faster, do not tend to activate the lymphatic system, and are removed from circulation slower [60]. BPPB showed positive ζ-p values (+18 mV), which was significantly different from that reported previously by Ceccacci et al. [36]. A positive surface charge is desirable for the development of highly effective nano-drugs. Based on the studies published so far, the internalization of positively charged NPs is more efficient than that of neutral and anionic NPs [61,62,63,64,65,66]. Notably, it was found that after electrostatic interaction with anionic components of the cell membrane as phospholipids, positively charged NPs can be internalized by several mechanisms, including pore formation, micropinocytosis, as well as clathrin- and dynamin-dependent endocytosis [62].

### 3.6. Antibacterial Properties

The antibacterial properties of BPPB were assessed first by determining its minimum inhibitory concentration (MICs) against several clinical isolates of both Gram-positive and Gram-negative species and then by performing 24 h time-killing experiments on representative species.

#### 3.6.1. In Vitro Antibacterial Activity of BPPB: Determination of MIC Values (MICs)

Here, for the first time, the synthetic C12 *bis*-triphenyl phosphonium salt (BPPB) has been investigated as a possible novel antibacterial therapeutic to treat severe human and/or animal infections sustained by MDR clinically isolated pathogens. In fact, while Wei et al., as well as Bin and Song, reported the sterilizing action of these types of compounds on environmental bacteria not connected to difficult-to-treat human infections, no evaluation has ever been carried out on MDR clinical strains so far [32,33]. Both studies were conducted on saprophytic bacteria (TGB), sulfate-reducing bacteria (SRB), and iron bacteria (IB) of no clinical relevance and suggested the sterilizing capacity of the compound at a high dosage of 20 mg/mL [32,33].

For MICs evaluation, 50 strains (49 MDR isolates and a fully susceptible *E. coli* of clinical origin) were exploited. Among enterococci, all strains were VRE, while staphylococci were all MRSA and MRSE. Three out of the six *S. epidermidis* also demonstrated resistance to linezolid. Among Gram-negative bacteria, except for *E. coli* 224, all isolates considered presented a complex pattern of resistance, including cross resistances to carbapenems, colistin, CAZAVI, and even the recently approved cefiderocol. We remember that colistin, or polymyxin E, is an older polycationic antibiotic often referred to as the “last-resort drug,” and traditionally employed in the management of Gram-negative bacterial infections, particularly those sustained by *Enterobacteriaceae* that have developed resistance practically to all other antibiotics [67]. Unfortunately, starting in 2016, many Gram-negative bacteria have been shown to possess genes that also confer resistance to colistin, thus further reducing the available weapons to treat the infections they cause [67]. Furthermore, cefiderocol is a novel strategic catechol-substituted siderophore cephalosporin. It binds to extracellular free iron and uses bacterial active iron transport channels to penetrate the periplasmic space of Gram-negative bacteria and kill them [68]. Nevertheless, cases of in vivo emerging cefiderocol resistance are increasingly being reported [69]. A variety of mechanisms of resistance have been reported, including β-lactamases (especially NDM, KPC, and AmpC variants conferring resistance to ceftazidime/avibactam, OXA-427, and PER- and SHV-type ESBLs) production, porin mutations, and mutations affecting siderophore receptors, efflux pumps, and target (PBP-3) modifications [69]. Against this background, it is of paramount importance and urgent to find novel compounds functioning on bacteria that have also developed resistance toward these last antibiotics. Moreover, *K. pneumoniae*, *K. aerogenes, E. cloacae*, *E. hormaechei* isolates, and four out of five *E. coli* were β-lactamase producing bacteria, including metallo-β-lactamase NDM and VIM-1, against which no antibiotic is functioning [70,71]. Two out of four *P. aeruginosa* were isolated from cystic fibrosis patients, while two out of four were pyoverdine- and pyocyanin-producing bacteria, which have been reported to be strains more tolerant to cationic antibiotics due to a repulsive action exerted by such cationic pigments [72]. Despite the worrying and complex scenario of resistance of the clinical strains tested in this study, BPPB has demonstrated powerful effects on everyone, regardless of their resistance strategy (Table 3 and Table 4).

Very low MICs, in the range 0.25–0.50 µg/mL (0.29–0.59 µM), were observed for all Gram-positive VRE, MRSA, and MRSE isolates considered here. These results demonstrate the high potential of BPPB to be developed as a new effective treatment for severe infections sustained by Gram-positive pathogens included in the ESKAPE group [22]. These are worrying Gram-positive and Gram-negative MDR pathogens, which have developed the capability to “escape” the traditional antibiotics, especially in the hospital setting. Particularly, VRE enterococcal species are MDR bacteria that, in addition to resistance to vancomycin, have already developed a variety of mechanisms of resistance to several other antibiotics like aminoglycosides, β-lactams, tetracyclines, and quinolones. Additionally, they can produce β-lactamases and have decreased cellular permeability, thus being the cause of severe hospital-acquired infections [73]. VREs are reported as the responsible number one for central line-associated bloodstream infections (CLABSI), number three for catheter-associated urinary tract infection (CAUTI), number eleven for ventilator-associated pneumonia (VAP), and number two for surgical site infections (SSI) [74]. Therefore, being BPPB active against these bacteria at a very low dosage (0.25–0.5 µg/mL) and considering the low cytotoxicity demonstrated in vitro against eukaryotic cells (see later in the paper) could represent the urgently required pharmacological tool necessary to limit infections by VRE, categorized as a “serious threat” by Centers for Disease Control (CDC) and Prevention [74]. Even more relevant is the effectiveness of BPPB against staphylococci, especially against MRSA, which ranks first in the USA in nosocomial infections, antibiotic-resistant pathogenic diseases, central line-associated bacteremia, and hospital-associated endocarditis [75,76]. Notably, MRSA is the first most common cause of community-acquired endocarditis in North America [77]. Very common in hospitals, prisons, and nursing homes, where immunocompromised patients and people with open wounds and/or invasive devices such as catheters are at greater risk of hospital-acquired infections, MRSA represents a global health threat and a clear ‘One Health’ problem. Moreover, MRSA can spread between and impact the environment, animals, and several human sectors [78].

Against MRSA, vancomycin is currently successful in approximately 49% of cases only, and its use is complicated by its inconvenient route of administration [79].

Unfortunately, several strains of MRSA have shown resistance to vancomycin and teicoplanin since the late 1990s [80]. Oxazolidinones such as linezolid, available from the 1990s, were initially beneficial in addressing resistance to vancomycin, but cases of bacteria tolerant to linezolid have been reported since 2001 [81].

However, for surgical site infections (SSIs) by MRSA [82] and for MRSA colonization in nonsurgical wounds such as traumatic wounds, burns, and chronic ulcers (i.e., diabetic ulcer, pressure ulcer, arterial insufficiency ulcer, and venous ulcer), no conclusive evidence has been found regarding the best antibiotic regimen to be used [83]. In this alarming scenario, made up of missing epidemiologic evidence, a plethora of uncertainties due to the interindividual responses of patients to existing antibiotics, and the decreasing efficacy of available drugs, the development of new curative options against MRSA infections is urgent. Therefore, another merit of the present study is that it has identified in BPPB a potentially new and strong antibacterial agent against MRSA, which is more active than most quaternary phosphonium salts developed thus far, as discussed in the following section. Higher MIC values were found on Gram-negative bacteria, but overall low (1–32 µg/mL (1.2–37.5 µM)) if we consider the very complex resistance model of strains used. Their distinctive structure makes them more difficult to be inhibited, than Gram-positive bacteria, thus being responsible of even high morbidity and lethal infections worldwide [84,85]. The highest MICs of 16–32 µg/mL were observed for MDR *K. pneumoniae* and *K. aerogenes* (16 µg/mL), *P. aeruginosa* (16–32 µg/mL), *E. cloacae* (16–32 µg/mL), and *E. hormaechei* (32 µg/mL) isolates. Lower MICs (4–8 µg/mL) were instead observed for MDR *A. baumannii* isolates, while MICs even lower than 1 µg/mL were found for MDR isolates of *E. coli* and *S. maltophilia,* regardless of their broad resistance and/or their resistance to carbapenems. Isolates of genera *Klebsiella, Escherichia,* and *Enterobacter* make part of the Gram-negative *Enterobacteriaceae* family [86]. Today, *K. pneumoniae* is the most common cause of hospital-acquired pneumonia in the United States, and the organism accounts for 3% to 8% of all nosocomial bacterial infections. It is noteworthy that the capability of *K. pneumoniae* to hydrolyze a very broad spectrum of β-lactam substrates, including penicillin, cephalosporins, monobactams, and carbapenems, is weakly inhibited by traditional β-lactam inhibitors, (i.e., clavulanic acid and tazobactam), and only the novel β-lactamase inhibitors (i.e., diazabicyclooctanes and boronates) are successful in contrasting isolates of this specie [70,71,87,88]. *K. aerogenes*, and other *Enterobacter* species, such as *E. cloacae* and *E. hormaechei* are responsible for bloodstream infections (BSI). BSI sustained by *K. aerogenes* and *E. cloacae* often presents poor clinical outcomes (death before discharge, recurrent BSI, and/or BSI complication), which are higher for *K. aerogenes* than *E. cloacae* [86]. In this regard, the pan-genome analysis revealed genes unique to *K. aerogenes* isolates, including putative virulence genes involved in iron acquisition, fimbriae/pili/flagella production, and metal homeostasis [86]. Regardless, antibiotic resistance was largely also found in *E. cloacae*. Isolates of genera *Pseudomonas, Acinetobacter,* and *Stenotrophomonas* make part of the Gram-negative non-fermenting bacteria. *P. aeruginosa* is commonly isolated from patients affected by both monomicrobial and polymicrobial infections [89]. The spectrum of infections caused by *P. aeruginosa* is wide, and the involvement of multiple organ systems is not uncommon [89]. To have available drugs active against these bacteria has become a significant problem and, to some extent, spurs the development of novel antimicrobial agents. Not less important, *Acinetobacter* species, mainly including *A. baumannii*, have developed resistance to multiple agents thus representing a common threat, as their infections are associated with substantial morbidity and mortality [89]. *S. maltophilia* is intrinsically drug-resistant to an array of different antibiotics and uses a broad arsenal to protect itself against antimicrobials. Surveillance studies have recorded a worrying increase in drug resistance for *S. maltophilia*, prompting new strategies to be developed against this opportunistic bacterium [90]. Although infection control and antimicrobial stewardship are important tools for combating the development and spread of infections caused by these bacteria, novel, appropriate antimicrobial therapies are urgently needed. In this scenario, thanks to this study, toward all these bacteria, which are intractable with currently available antibiotics and responsible for serious life-threatening infections, it will be possible to hypothesize a future use of the BPPB developed here.

##### An Overview of Previous Achievements by Using Antibacterial Phosphonium Salts

Ermolaev et al. synthesized a series of 20 sterically hindered quaternary phosphonium salts having one cationic head, which were tested on selected Gram-positive and Gram-negative bacteria and fungi, including, among others, strains of *E. faecalis*, MRSA, *E. coli* and *P. aeruginosa*, also considered by us [1]. According to the MICs reported by the authors for these isolates, when tested on several MRSA and VRE *E. faecalis* strains, BPPB was demonstrated to be more potent than all compounds developed and tested by Ermolaev [1]. Additionally, when ciprofloxacin was tested by the authors, it demonstrated MICs higher than those of BPPB by 1.8–500 times against MRSA and by 7.8–15.6 times against an *E. faecalis* isolate, whose pattern of resistance has not been clearly reported [1]. Pugachev and colleagues synthesized 13 pyridoxine-based phosphonium salts and tested in vitro their antibacterial effects on Gram-positive isolates of staphylococcal genus and on strains of *K. pneumoniae* and *Proteus* spp. genus as representative isolates of Gram-negative species [30]. No compound that authors tested on their bacteria as a reference antibiotic, including vancomycin, was active on Gram-negative bacteria (MICs > 1000 µg/mL), while the best performant molecule (compound 20) demonstrated MICs = 5 µg/mL against both *S. aureus* and *S. epidermidis*, by cell penetration and interaction with genomic and plasmid DNA [30]. Based on these results, BPPB was 10-20-fold more potent than compound 20 developed by Pugachev et al. against MRSA and MRSE [30]. When vancomycin was used on *S. aureus* and *S. epidermidis* by the authors in their study, MICs were 5–10 times higher than those displayed by BPPB against MRSA and MRSE [30]. In the same year, phosphonium salts with two cationic heads, like BPPB, were synthesized by Pugachev et al. Particularly, the authors produced 23 novel *bis*-phosphonium salts based on pyridoxine and their antibacterial activities were evaluated *in vitro*. As in their previous study, despite the presence of two cationic heads, all compounds were inactive against Gram-negative bacteria, including *K. pneumoniae* isolates (MICs > 1000 µg/mL). On the contrary, BPPB displayed low MICs = 16 µg/mL on all *K. pneumoniae* isolates tested, regardless of their resistance to carbapenems, their capability to produce of KPC and/or VIM-1 β-lactamase, and/or their resistance to cefiderocol. The compounds developed by Pugachev et al. exhibited structure-dependent activity against Gram-positive isolates [34]. Anyway, in the best cases (compound 14a and 10a in the study), MICs against *S. aureus* and *S. epidermidis* with not specified resistance were 1–1.25 µg/mL, thus establishing that BPPB was 2-5-fold more potent than 10a and 14a, regardless of the MDR MRSA and MRSE strains used by us [34]. In the past, Cieniecka-Roslonkiewicz and co-workers synthesized 21 alkyl phosphonium salts and essayed them against a selection of Gram-positive and Gram-negative ATCC bacteria. From the reported results, the best performant compounds prepared by the authors were 4-8-fold less active than BPPB when used on MRSE, 4.6–9.2 less potent than BPPB when tested against MRSA, and 4.6–9.2 less potent than BPPB when tested against VRE *E. faecium* [31]. Additionally, when benzalkonium chloride was used by the authors on ATCC strains of *S. epidermidis*, *S. aureus*, and *E. faecium*, MICs 2.8-5.6-, 5.6-11.2-, and 11.2-22.4-times higher than those shown by BPPB against MDR isolates used by us were detected [31]. More recently, Lei et al. synthesized two mono- and one *bis*-quaternary phosphonium tosylate compounds with different lengths of oligo-(ethylene glycol) (OEG) chains and TPP^+^ moieties as an antibacterial group [35]. The aim of the authors was to find new antibacterial compounds with less toxicity with respect to TPP-based smaller molecules previously reported. As expected, when synthesized compounds were essayed on *S. aureus* and *E. coli*, the *bis*-quaternary phosphonium salt was remarkably more active than mono ones, but the lowest MICs recorded were 1.5 mg/mL on *S. aureus* and 3.1 mg/mL on *E. coli*. Collectively, although when administered at concentrations up to 2.5 mg/mL (only 1.7× MIC) to eukaryotic cells, viable cells were almost 80% respect to control after 3 days, the doses necessary to inhibit *S. aureus* and *E. coli* ATCC strains were 3000-6000-fold and 1550-3100-fold higher respectively, than those of BPPB, which were necessary to inhibit MRSA and MDR *E. coli* isolates. BPPB was also more effective than a phosphonium salt recently reported by us (compound 1 in the study) and tested on Gram-positive and Gram-negative isolates with complex patterns of resistance like that of the strains used in this study [13]. Completely inactive against Gram-negative species, 1 was 8–32 times less active than BPPB against staphylococci and 16–64 times less active against enterococci [13]. Very recently, Nunes and co-workers synthesized seven C6-C18 alkyl triphenyl phosphonium salts from a library of 49 QASs and QPSs and evaluated their antibacterial effect on antibiotic-sensitive and antibiotic-resistant *S. aureus* [85]. Collectively, salts with C12-C14 alkyl chains were the best performant antibacterials, with phosphonium salts being significantly more efficient than ammonium compounds. When tested on antibiotic-resistant *S. aureus*, C14 alkyl triphenyl phosphonium salt (1e) displayed MICs of 1–2 µg/mL, thus establishing that BPPB was more potent by 2–8 times. On these resistant strains, erythromycin, tetracycline, and ciprofloxacin, tested by the authors for comparison, displayed MICs 512-1024-fold, 256-512-fold, and 256-512-fold higher than those demonstrated by BPPB on eight MDR MRSA [85]. Also, when eight new chiral quaternary phosphonium salts (CQPSs) comprising (1R,2S,5R)-(−)-menthyl or (1S,2R,5S)-(+)-menthyl groups were tested by Arkhipova and colleagues on Gram-positive MRSA-1 and MRSA-2 strains, sensitive *E. faecalis*; as well as on *E. coli* and *P. aeruginosa,* as Gram-negative pathogenic bacteria, they were all inactive against the latter species. Better activity was observed against MRSA and *E. faecalis*, but the best performant compound was 15.6-31-fold less active than BPPB when was examined against MRSA and VRE *E. faecalis* [48]. From literature data about the antibacterial performance of previously reported phosphonium salts, it was possible to compare the antibacterial properties of BPPB with those of previously reported similar compounds only concerning *E. coli, P. aeruginosa*, and/or *K. pneumoniae* because the other clinically relevant species assayed in this study, have not been considered by other authors which handled phosphonium salts. BPPB demonstrated antibacterial effects against MDR *E. coli* and *P. aeruginosa*, comparable or higher than those showed by sterically hindered quaternary phosphonium salts having one cationic head on sensitive strains of the same genus [1]. Differently from our BPPB, all the 21 alkyl phosphonium salts developed by Cieniecka-Rosłonkiewicz were not active against *P. aeruginosa*, while the best performant compound was 1.25-2.50-fold less active than BPPB on an isolate of *E. coli* with not specified pattern of resistance [31]. When benzalkonium chloride was used on *E. coli* and *P. aeruginosa* tested in the study by Cieniecka-Rosłonkiewicz, it demonstrated MICs 1.4-2.8-times and 5.5-11-times higher than those displayed by BPPB against isolates of the same genus, but with complex resistance profiles [31]. A very recent interesting study by Summers and co-workers reported the synthesis, characterization, and antibacterial properties of 59 quaternary phosphonium compounds (QPCs), including mono-QPCs, bis-QPCs, tris-QPCs, and tetra-QPCs, bearing 1 to 4 C8-C18 hydrocarbon tails [91]. Their antibacterial activity was essayed against two strains of MRSA, one MSSA, one *E. faecalis*, one *E. coli,* and one *P. aeruginosa*, evidencing MICs in the ranges 0.5–125 µM on MRSA, 0.5–16 µM on MSSA, 2->250 µM on *E. faecalis*, 2->250 on *E. coli* and 4->250 on *P. aeruginosa.* Based on these results, BPPB was more potent than all compounds reported in a study on MRSA, *E. faecalis*, and *E. coli* [91]. Additionally, although 13 of the compounds reported by Summers et al. were more active than BPPB on *P. aeruginosa* used in their study, *P. aeruginosa* isolates used by us presented a much more complex pattern of resistance [91].

#### 3.6.2. Time-Killing Curves

To assess whether BPPB was bactericidal or bacteriostatic, time-kill experiments were performed at concentrations equal to 4× the MIC for strains of MRSA and *E. coli*. Particularly, isolates 17, 18, 187, and 195 of MRSA and strains 477, 525, and 539 of *E. coli* were used. As depicted in Figure 12, reporting the most representative curve obtained for MRSA strain 18 and *E. coli* 539, BPPB showed bacteriostatic effects against both species as a decrease of <3 log in, the original cell number was evident after 6 h of exposure. During the next 24 h, a slight increase and regrowth occurred.

A series of quaternary phosphonium (QP) N-chloramine molecules, covalently combining an N-chloramine and a QP moiety via aliphatic chains made of different methylene units, were synthesized using multi-step chemical reactions [92]. Compared with a quaternary ammonium salt synthesized by the authors as a reference compound, QP compounds displayed significantly higher antibacterial effects and were biocidal depending on the time of exposure [92]. While the QP N-chloramine compounds with C3–C12 chains as linkers (compounds **3**–**6**) demonstrated high biocidal efficacy after 10 min of contact, causing a 6.22 and 7.30 CFU log reduction in the initial inoculum of *E. coli* and *S. aureus,* respectively, compound 6 bearing a C12 chain was bactericidal after only 5 min of contact [92]. This suggests that future chemical modification of BPPB by inserting N-chloramine residues could be a strategy to convert the bacteriostatic effects observed for BPPB into bactericidal effects.

### 3.7. Cytotoxicity Experiments on Eukaryotic Cells

The assessment of cytotoxicity toward eukaryotic cells is essential to predict the possible development of new compounds as antimicrobial agents suitable for clinical application. As reported, the cytotoxicity of QPSs could depend on their different physicochemical characteristics, such as the length of alkyl chains, the presence of aromatic ring(s), on which the P^+^ cation can delocalize, thus conferring the molecules a softer charge [36], the type of anion, the possible colloidal properties, and the capability or not to form nano aggregates [85]. Additionally, QPSs cytotoxicity could only depend on the specific cell type under investigation [85]. Concerning this, bacteria and cancer cells present differences from mammalian cells. The presence on the bacterial surface of lipids with a net anionic charge in place of zwitterionic lipids neutral at physiological pH present in that of mammalian cells translates into a significant disparity in membrane electrostatic charges, thus resulting in the targeted tropism of cationic antimicrobial agents toward negatively charged bacteria [27]. Additionally, cholesterol present in mammalian cell membranes and missing in those of bacteria could provide selective protection to eukaryotic cells, thus reducing the possible cytotoxicity of a novel antibacterial compound upon exposure. Regardless, depending on the concentration, time of exposure, and the abovementioned structural properties of QPSs, cytotoxicity may still manifest in mammalian cells [85]. In this study, the cytotoxicity profile of BPPB was assessed in both human liver cells (HepG2) cells as in our previous work [13] and in transformed African green monkey kidney fibroblast cells (Cos-7). HepG2 cells are commonly employed in drug metabolism and hepatotoxicity studies [93,94,95], while Cos-7 is commonly used in toxicology research and gene-transfection experiments. Since HepG2 cells exhibit a non-tumorigenic nature, display high proliferation rates, and possess an epithelial-like morphology while retaining many differentiated hepatic functions [94,96], they represent a useful model for evaluating the cytotoxic effects of novel compounds. A sufficiently high value of the selectivity index (SI) is an essential requirement to render a new molecule worthy of consideration for further studies and future development as a new therapeutic agent. The SI value is given by the equation Equation (1) and provides a measure of the selectivity of the antimicrobial agent for bacteria.
*SI* = *IC*_50_/*MIC*(1)

In Equation (1), IC_50_ is the concentration of a compound capable of reducing viable cells by 50%, while MIC is the minimum inhibitory concentration determined for a specific isolate. To obtain the SI values of BPPB, we carried out dose–dependent experiments using the MTT cell proliferation tests. Additionally, we evaluated the cytotoxicity of BPPB using a lactate dehydrogenase (LHD) cytotoxicity assay, which measures LDH activity as an indicator of cell death. The results from the MTT test were used to calculate the IC_50_ values for both cell lines, and the obtained IC_50_ and MICs, which were observed most frequently for each bacterial species considered (Table 3 and Table 4), were then used to calculate the SI values.

#### 3.7.1. Assessment of the BPPB Concentration-Dependent Effects on Cos-7 and HepG2 Cells by MTT and LDH Essays

Both cell lines were exposed for 24 h to different concentrations of BPPB 0.5–100 µg/mL (0.0 µg/mL was the control) on the basis of the observed MICs. Growth inhibition/death was determined by the MTT test, while cell cytotoxicity was determined by the LDH assay. The time of 24 h of treatment was chosen to have data directly comparable with those obtained in MICs, which must be determined after 24 h as for the EUCAST indications [42]. The results are reported as bar graphs in Figure 13 and dispersion graphs in Appendix A.

Particularly, in the case of the MTT test (Figure 13a,b), bar graphs show the cell viability (%) of Cos-7 (**a**) and HepG2 (**b**) cells untreated (CTRL) and after exposure to increasing concentrations of BPPB for 24 h, while in Figure 13c,d, we have plotted the quantification of cell damage (expressed as optical density (OD) units of absorbance)) of Cos-7 (**c**) and HepG2 (**d**) cells untreated (CTRL), and after exposure to increasing concentrations of BPPB for 24 h. Moreover, representative phase contrast images of Cos-7 (**e**) and HepG2 (**f**) cells acquired in untreated conditions or after exposure to increasing concentrations of BPPB for 24 h have been reported.

According to the results shown in Figure 13a,b, in the MTT test, BPPB demonstrated concentration-dependent effects on both cell lines. However, for concentrations = 10 µg/mL, while on HepG2 cells, BPPB exerted effects similar to those observed at 5 µg/mL, and cell viability was higher than 50% and similar (64.5% vs. 62.9%) at both concentrations, the cell viability of Cos-7 decreased under 30% (27.45%) compared to that observed at 5 µg/mL (64.2%). Collectively, the MTT test demonstrated that cell viability was affected by BPPB exposure to a greater extent in Cos-7 cells than in HepG2 cells, and a safety profile was detected at concentrations ≤ 5 µg/mL toward Cos-7 and ≤10 µg/mL toward HepG2 cells, which is, however, a scenario more satisfactory than that reported by Nunes et al., for the best performing antibacterial compounds they recently synthesized [85]. From the LDH test results (Figure 13b,c), a slightly different scenario was evidenced, in which, while a dose–dependent effect was observed at concentrations ≥ 10 µg/mL on Cos-7 cells and similar cell damage was observed for concentrations ≤ 5 µg/mL, on HepG2 cells a dose–dependent effect was observed at concentrations ≤ 10 µg/mL and similar cell damage was observed for concentrations ≥ 50 µg/mL. Nonetheless, as observed in the MTT test, for concentrations = 10 µg/mL, while on HepG2 cells, BPPB caused damage like that observed at 5 µg/mL and the optical density due to LDH release was similar (OD = 0.71 vs. 0.79) at both concentrations, that measured on Cos-7 cells remarkably increased (OD = 0.80) with respect to that observed at 5 µg/mL (OD = 0.30), thus evidencing remarkably higher cell damage. Collectively, the LDH test also demonstrated that a safety profile exists at concentrations ≤ 5 µg/mL for Cos-7 and ≤10 µg/mL for HepG2 cells.

#### 3.7.2. IC_50_ and Selectivity Indices

For a more precise evaluation of the effects of BPPB on eukaryotic cells, we calculated the IC_50_ of BPPB in both cell lines using GraphPad Prism software and fitting data with nonlinear regression models. Particularly, we plotted the Log_10_ of the concentrations vs. the values of cell viability observed, and the fitting nonlinear models were achieved (Appendix A). The calculated IC_50_ values were 5.76 ± 0.95 µg/mL and 11.31 ± 1.54 µg/mL on Cos-7 and HepG2 cells respectively, thus confirming the significantly dissimilar cytotoxicity of BPPB on the different cells, being Cos-7 the cells less tolerant to BPPB and confirming the previously reported assertion that the toxicity of a new compound strongly depends on the type of cell line used [85]. In this regard, considering that Cos-7 cells are derived from a non-human source, the results obtained using Cos-7 cells may not be directly applicable to human biology. So, the lower toxicity of BPPB found toward human hepatic cells with respect to monkey kidney ones and not the contrary is a very promising result, which could pave the way for the future clinical development of BPPB and its possible derivatives. Additionally, since it is believed that the HepG2 cell line retains most of the metabolic functions of normal hepatocytes, and it is commonly used to study the toxic effects of heavy metals, nanoparticles, and drugs in vitro [97], we retained more reliable cytotoxicity results obtained on HepG2 than those obtained on Cos-7 cells. Anyway, to assess the bacterial species against which BPPB could be though as a new promising therapeutic treatment, using Equation (1), the values of IC_50_ from the MTT test, and the MICs observed with most frequency for all bacterial species reported in Table 3 and Table 4, the SIs of BPPB were calculated for both cell lines. The results have been included in Table 5 for Gram-positive isolates and Table 6 for Gram-negative isolates.

For prospective antimycobacterial drugs, it has been proposed that the selectivity index should be greater than ten before they are considered for further development [98]. Accordingly, and according to findings reported in Table 5 and Table 6, and considering the most reliable results, those achieved on HepG2 cells, BPPB could be considered very promising as an effective agent against the most worrying MDR bacterial strains of Gram-positive species tested here and *E. coli*, thus providing valuable insights for the rational design of other BPPB like derivatives with enlarged SI values.

## 4. Conclusions

In this study, a new potent antibacterial agent possessing bacteriostatic effects against MDR clinical isolates of Gram-positive and Gran-negative species, including ESKAPE bacteria, was developed. By performing a low-cost one-step reaction, a *bis*-triphenyl phosphonium di-bromide salt bearing a C12 alkyl chain as the linker between the two cationic heads (BPPB) was prepared and fully characterized. Known for its colloidal properties and sterilizing action toward environmental bacteria not involved in severe human infections needing novel treatments urgently, BPPB has never been tested on clinically relevant MDR bacteria against which currently available antibiotics fail. Here, for the first time, the antibacterial activity of BPPB was investigated in fifty MDR clinical isolates of both Gram-positive and Gram-negative species responsible for serious human infections that are difficult to manage due to the complex pattern of resistance of pathogens. Results from MICs determination established that BPPB is active at low to very low MICs against all the species tested, regardless of their worrying resistance. Moreover, it caused a 2 log CFU reduction in the initial inoculum after 6 h of exposure in 24 h time-killing experiments. The antibacterial activities of BPPB have been compared with that of several QPSs previously reported establishing the superiority of BPPB in almost all cases. Moreover, for the first time, the cytotoxicity of BPPB was investigated in human liver cells HepG2 and monkey kidney cells Cos-7 by carrying out MTT and LHD assays. Both assays showed similar patterns of toxicity and safety profiles at concentrations ≤ 5 µg/mL for Cos-7 and ≤10 µg/mL for HepG2 cells. Considering results obtained on HepG2 cells which are a better in vitro model for human biology, very high selectivity indices (SIs) > 10 were obtained against all MDR Gram-positive isolates considered and *E. coli,* while SIs = 1.4–5.6 were detected also on worrying Gram-negative species such as *A. baumannii* and *S. maltophylia*. This study also confirmed that antibacterial quaternary phosphonium salts (QPSs) could be more effective and less cytotoxic than their ammonium counterparts (QASs), that *bis*-QPSs could be more potent antibacterials than mono-QPSs, that hindered QPSs such as TPP-based compounds could be more active and less cytotoxic than other QPSs, and that bola-amphiphile TPP-based molecules possess peculiar colloidal properties allowing them to self-assemble in nanomicelles, thus improving their biological effect, as often observed for nanomaterials. Collectively, BPPB could therefore represent a novel potent template molecule to develop clinically applicable new antibacterial devices for counteracting infections sustained by antibiotic-resistant superbugs that are not manageable with currently available antibiotics. Future improvements in the antibacterial effects of BPPB while reducing its cytotoxic profiles could be derived by varying the length of the alkyl chain that acts as a linker between the two cationic heads and, therefore, by the liposomal formulation of the most promising compounds using themselves as cationic lipids.

## Data Availability

All necessary data are included in this manuscript and in the related Appendix A.

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
