# Peer review of "Synthesized Bis-Triphenyl Phosphonium-Based Nano Vesicles Have Potent and Selective Antibacterial Effects on Several Clinically Relevant Superbugs"

_nanomaterials, 2024, doi:10.3390/nano14161351_

Round 1

Reviewer 1 Report

Comments and Suggestions for Authors

In the manuscript “Synthesized Bis-Triphenyl Phosphonium-Based Nano Vesicles Have Potent and Selective Antibacterial Effects on Several Clinically Relevant Superbugs” by Alféizar et al. the antibacterial activity of a new phosphonium bolaform surfactant, BPPB, against several Gram-positive and Gram-negative bacteria was investigated. This BPPB shows a lower toxicity than previous antibiocides. The results seem to point out that BPPB could be promising for being clinically developed. The manuscript is well written and the results are interesting, the conclusions being suported by the experimental data. However, a large part of the text correspond to results previously obtained (see ref. 38). The synthesis and characterization should be deleted, this means most of section 2 and the beginning of section 3. Besides, the Introduction is too long. That is, the manuscript could be published in Nanomaterials after a major revision. The text has to be rewritten.

Author Response

In the manuscript “Synthesized Bis-Triphenyl Phosphonium-Based Nano Vesicles Have Potent and Selective Antibacterial Effects on Several Clinically Relevant Superbugs” by Alféizar et al. the antibacterial activity of a new phosphonium bolaform surfactant, BPPB, against several Gram-positive and Gram-negative bacteria was investigated. This BPPB shows a lower toxicity than previous antibiocides. The results seem to point out that BPPB could be promising for being clinically developed. The manuscript is well written and the results are interesting, the conclusions being suported by the experimental data. However, a large part of the text correspond to results previously obtained (see ref. 38). The synthesis and characterization should be deleted, this means most of section 2 and the beginning of section 3. Besides, the Introduction is too long. That is, the manuscript could be published in Nanomaterials after a major revision. The text has to be rewritten.

Sorry, simply my name is Alfei and not Alféizar, but it is of minimal importance. Most important, we thank a lot the Reviewer for his/her positive comments. Collectively, most text, especially the Introduction section has been rewritten as asked by the Reviewer. Anyway, although his/her suggestion to delete the synthesis and characterization of BPPB because it could seem correspondent to what reported by Ceccacci et al in Ref. 38 (original version, now Ref. 37) may appear reasonable, it would cut off essential contents for the present paper and readers, because this part provides essential advancements and news. In fact, only the structure of the compound reported in Ref. 37 corresponds to that of BPPB reported here, while both the synthetic procedure and characterization differ and provided new interesting findings. If the Reviewer would better consider the synthesis and characterization reported previously by Ceccacci et al (Ref. 37) and that reported here by us, she/he would realize that they differ significantly, and therefore we believe it is necessary to report them in the experimental part, as well as in the discussion section. Concerning the synthesis of BPPB, we use EtOH as solvent in place of DMF, which adjuvated the solvent removal at the end of reaction, thus providing the final product as a solid after a single crystallization, without need of all the operations reported by Ceccacci. Ph3P was used exactly in 2/1 ratio with the dibromide, thus avoiding residual reagent to be removed (as reported by Ceccacci), who had to perform a more complicated work up to isolate the final product, with major waste of solvents. We experimented that at room temperature (as Ceccacci reported), the reaction did not reach the completion. So, we conducted the reaction at reflux for longer time, but without using inert atmosphere, thus reducing the production costs. Concerning BPPB characterization, differently from Ceccacci et al, who acquired the NMR spectra in CD3OD, we acquired them in CDCl3, thus observing more and significantly different signals and more coupling. We reported also DEPT-135 experiments, the 31P NMR spectrum and ATR-FTIR analyses not described previously. The melting point of BPPB and its elemental analyses, not reported in Ref. 37 were also furnished by us. We provided for the first time also the images of all spectra acquired and a detailed assignation of peaks and bands. The behaviour in solution of the bola-amphiphile (BPPB) prepared by us was different from that of compound 1 previously reported, providing (upon correct dilution) only one-dimensional family of smaller size (49 nm) with a lower positive Z-potential, which probably limited its cytotoxicity on eukaryotic cells. Also, the UV-Vis spectrum carried out by us evidenced more intense additional peaks at lower nm (205 and 224 nm) not detected by Ceccacci et al. Potentiometric titrations not provided in Ref. 37 concluded the BPPB characterization performed by us. We think that all these reasons could justify the utility of sections 2.2-2.6 and 3.1-3.5 in the present paper. Anyway, in addition to those already present, some sentences have been included in the main text to better evidencing these facts. Please, see lines 240-241, 267-268, 426-441, 484-502. Regarding the second concern of the Reviewer about the introduction, as above mentioned, it has been extensively shortened. Anyway, we have been forced to add a new part on request of the second Reviewer.

Reviewer 2 Report

Comments and Suggestions for Authors

The manuscript entitled: “Synthesized Bis-Triphenyl Phosphonium-Based Nano Vesicles Have Potent and Selective Antibacterial Effects on Several Clinically Relevant Superbugs” by Silvana Alfei and coworkers has been revised. In this paper, the antibacterial effects of BPPB were for assessed against fifty clinical isolates of both Gram-positive and Gram-negative species, where excellent antibacterial effects were observed on all strains tested. Specifically, in HepG2 human hepatic and Cos-7 monkey kidney cell lines, BPPB showed selectivity indices for all Gram-positive isolated and for clinically relevant Gram-negative superbugs such as those of E. coli species, which have important clinical implications. The manuscript is well-written and sounds interesting for the scientific community of Nanomaterials journal. However, some aspects of the manuscript should be improved and I feel that the paper needs a major revision before it can be published in Nanomaterials.

 -In the introduction section the important role of gold and silver gold nanoparticles and gemini surfactants in combating antibiotic resistance should be clarified and included in the text.

- Figures 4-6 and 11 are with low quality. Please prove the pictures with high resolution.

-The size of vesicles in Figure 11 is different to the size obtained by optical microscopy in Figure 10. Please, justify the reason for this difference. Moreover, the results in Figure 11 can also be specified in size-number type curve.

-Results in figures 13 e-f are highly important for justify the conclusion given in the work. However, the cells are not properly visualized. Why 24 hours of treatment was selected instead 48 hours? Is that means the result of the treatment not varied?

-How to calculate the cell viability percentage? Please specify. Has any type of correction been made to the absorbance data in Figures 13 c and d?

Comments on the Quality of English Language

In my opinion only minor editing is required. 

Author Response

The manuscript entitled: “Synthesized Bis-Triphenyl Phosphonium-Based Nano Vesicles Have Potent and Selective Antibacterial Effects on Several Clinically Relevant Superbugs” by Silvana Alfei and coworkers has been revised. In this paper, the antibacterial effects of BPPB were for assessed against fifty clinical isolates of both Gram-positive and Gram-negative species, where excellent antibacterial effects were observed on all strains tested. Specifically, in HepG2 human hepatic and Cos-7 monkey kidney cell lines, BPPB showed selectivity indices for all Gram-positive isolated and for clinically relevant Gram-negative superbugs such as those of E. coli species, which have important clinical implications. The manuscript is well-written and sounds interesting for the scientific community of Nanomaterials journal. However, some aspects of the manuscript should be improved and I feel that the paper needs a major revision before it can be published in Nanomaterials.

 -In the introduction section the important role of gold and silver gold nanoparticles and gemini surfactants in combating antibiotic resistance should be clarified and included in the text.

We thank the Reviewer for his/her suggestion which allowed us to make the Introduction of our paper more complete, by reporting on the antibacterial use of metallic NPs and gemini ammonium salts. Please, see lines 66-69 and 103-106.

- Figures 4-6 and 11 are with low quality. Please prove the pictures with high resolution.

As asked, the quality of Figures 4-6 and 11 has been improved. Anyway, separate images with the resolution required by Nanomaterials have been uploaded in a zipped folder during the submission procedure.

-The size of vesicles in Figure 11 is different to the size obtained by optical microscopy in Figure 10. Please, justify the reason for this difference. Moreover, the results in Figure 11 can also be specified in size-number type curve.

The answer to the Reviewer question was already included in the original version of the manuscript. Please, reconsider Sections 3.4 and 3.5 and Figures S2-S4 in the Supplementary Materials file. As reported in Section 3.4., optical microscopy is not the best method to assess the morphology and dimensions of particles, in fact we use it for an early investigation, only, followed by the more suitable DLS. Sensibility of optical microscopy is limited and to be sure to see something, highly concentrated solutions were needed. In fact, as reported in the main text, when optical images were captured, a high concentration solution tending to give an opaque suspension on cooling (4.4-fold higher the CAC) was prepared to overcome the limited performances of the instrument. This caused the formation of large aggregates with dimensions remarkably higher than those provided by DLS analyses, when carried out on diluted and filtrated clear solutions. On the contrary, for the DLS analyses a lower concentration solution was used which evidenced more than one dimensional family, among which one reproduced the dimensions observed in the optical images. Upon dilution and filtration, we obtained the result reported in Figure 11.  Additional specification on the question have been included in the abstract (lines 30-33). For the second request from the Reviewer, we make kindly him/her note that the size-number type curve is a mere mathematical transformation of the size-intensity one naturally provided by the DLS instrument. The use of these curves, in some cases, could provide very misleading outcomes and lead to erroneous conclusions, which we prefer aviding. Please, consider https://www.malvernpanalytical.com/en/learn/knowledge-center/insights/intensity-volume-number-which-size-is-correct/ .

-Results in figures 13 e-f are highly important for justify the conclusion given in the work. However, the cells are not properly visualized. Why 24 hours of treatment was selected instead 48 hours? Is that means the result of the treatment not varied?

We agree with the Reviewer that visualization of cells is not optimized but having used a 96 multi-wells plate in which wells are very small, the images of cells cannot be further improved. We kindly ask the Reviewer to accept the reported images as such. The time of 24 hours of treatment was chosen to have data directly comparable with those obtained in MICs determination, which must be measured after 24 hours as for the EUCAST indications (Ref. 43). Sentences to specify this question has been now included in the main text (lines 372-373, 399 and 941-943)

-How to calculate the cell viability percentage? Please specify. Has any type of correction been made to the absorbance data in Figures 13 c and d?

 The answer to the first question of the Reviewer was already reported in section 2.8.1. of the original manuscript. Please consider the sentence “The cell survival rate, expressed as cell viability percentage (%), was evaluated based on the experimental outputs of treated groups vs. the untreated groups (CTR) and was calculated as follows: cell viability (%) = (OD treated cells − OD blank)/(OD untreated cells − OD blank) × 100%.” Concerning the second question, please consider the other sentence already present in te original manuscript: “Positive control outputs were used as the maximum LDH signal. The LDH content (OD at 450 nm) quantified in the cultured medium collected from treated or untreated conditions is directly proportional to damaged cells.

Comments on the Quality of English Language

In my opinion only minor editing is required.

With the help of our colleague prof Deirdre Kantz, teacher mother tongue of English at University of Genoa and Pavia, the work has been revised to reduce typos and grammatical errors.